# SYMBOLIC VARIABLES IN DISTRIBUTED NETWORKS THAT COUNT

**Satchel Grant**
Department of Psychology
450 Jane Stanford Way
Stanford, CA 94305 USA
`grantsrb@stanford.edu`

**Zhengxuan Wu**
Department of Computer Science
450 Jane Stanford Way
Stanford, CA 94305 USA
`wuzhengx@stanford.edu`

**James L. McClelland**
Department of Psychology
450 Jane Stanford Way
Stanford, CA 94305 USA
`jlmcc@stanford.edu`

**Noah D. Goodman**
Department of Psychology
Department of Computer Science
450 Jane Stanford Way
Stanford, CA 94305 USA
`ngoodman@stanford.edu`

## ABSTRACT

The discrete entities of symbolic systems and their explicit relations make symbolic systems more transparent and easier to communicate. This is in contrast to neural systems, which are often continuous and opaque. It is understandable that psychologists often pursue interpretations of human cognition using symbolic characterizations, and it is clear that the ability to find symbolic variables within neural systems would be beneficial for interpreting and controlling Artificial Neural Networks (ANNs). Symbolic interpretations can, however, oversimplify non-symbolic systems. This has been demonstrated in findings from research on children's performance on tasks thought to depend on a concept of exact number, where recent findings suggest a gradience of counting ability in children's learning trajectories. In this work, we take inspiration from these findings to explore the emergence of symbolic representations in ANNs. We demonstrate how to align recurrent neural representations with high-level, symbolic representations of number by causally intervening on the neural system. We find that consistent, discrete representations of numbers do emerge in ANNs. We use this to inform the discussion on how neural systems represent exact quantity. The symbol-like representations in the network, however, evolve with learning, and can continue to vary after the neural network consistently solves the task, demonstrating the graded nature of symbolic variables in distributed systems.

## 1 INTRODUCTION

Both biological and artificial Neural Networks (NNs) have powerful modeling abilities. Aside from the impressive capabilities of human cognition in biological NNs, the more recent artificial NNs (ANNs) have had such great success that they have been crowned the "gold standard" in many machine learning communities (Alzubaidi et al., 2021). Despite their success, however, the inner workings of NNs remain largely opaque to humans, partly because representations in NNs are often highly distributed. Individual neurons in NNs can play multiplex roles within a single network (McClelland et al., 1986; Smolensky, 1988). Meanwhile, current tools often lack the ability to uncover precise mechanisms of ANNs from their distributed representations.

Symbolic systems, in contrast, defined by clear, discrete entities and explicit rules and relations, have the benefit of greater interpretability. These systems can often be designed with the goal of maintain-

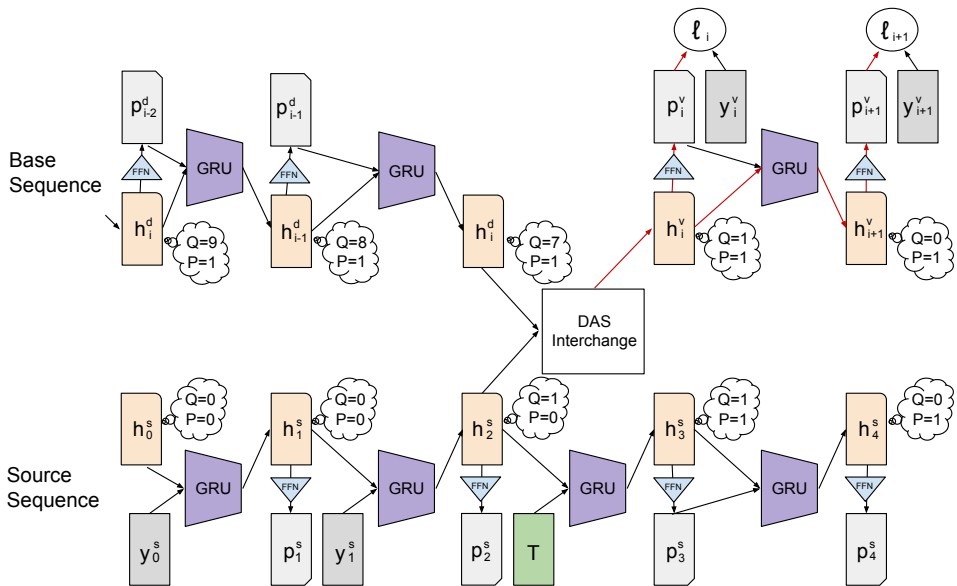

Figure 1: Diagram of a causal interchange performed on the hidden state of the GRU in the base sequence. In general, the triangles are Feed Forward Networks (FFNs); the trapezoids are GRU cells; the thought clouds show the values of our hypothesized symbolic variables where $Q$ is the Quantity variable and $P$ is the Phase variable; in the circles in the upper right corner, $\ell_i$ and $\ell_{i+1}$ denote cross entropy loss over predicted logits; and all other shapes other than the DAS Interchange rectangle are either $h$, for hidden state vector; $p$ for predicted logits; or $y$ which is a one hot encoding of ground truth token. During the demonstration phase of the task, ground truth tokens are fed into the GRU. During the response phase, we use the models' predictions as input to each subsequent step in the GRU. The trigger token $T$ signals the phase transition from demonstration to response. For simplicity, we draw an arrow directly from the logits or one hot encodings into the GRU cell, but in actuality, we use an embedding selected from the unit with the maximum value. We harvest variables from the source sequence and put them into the destination sequence for the causal interchange. In this example the source hidden state is $h_2^s$ and the destination hidden state is $h_i^d$ where the subscript denotes the index in the sequence, and the superscript denotes what sequence it is used in—$s$ for source, $d$ for destination (referred to as "base" in previous work), and $v$ for intervened. We use a trained model with frozen weights to produce all predictions. We back-propagate into the DAS Interchange rotation matrix using prediction error on the expected sequence under the hypothesized symbolic variables and causal interchange. Red arrows indicate the gradient flow in back-propagation.

ing mechanistic simplicity and interpretablity. This usually comes with the benefit of causal power over the systems, enabling us to change intermediate components for a desired output. Symbolic systems can lack, however, the expressivity and performance capabilities of NNs. This is apparent in the field of natural language processing where NNs such as Transformers (Vaswani et al., 2017) have swept the field. The field has witnessed a transformation from the power of scalable learning objectives and model scale using ANNs (Brown et al., 2020; Kaplan et al., 2020), easily surpassing the previously existing, purely symbolic approaches. Is it possible to gain the interpretability of a symbolic system in these NNs? How can we explore the causal relationships of NN representations?

Regardless of the performance differences of symbolic vs neural models, a main goal of scientific discovery is centered on the generation of simplified, symbolic interpretations of complex systems. These simplifications are necessary for understanding the essential parts of a system and how they causally interact. This form of understanding grants goal directed agents causal power over the system. Indeed, it could be argued that symbolic simplifications are necessary for goal directed agents to successfully interact with the world, or at least to share information with each other about how to do so.

One common symbolic system is the system of natural or counting numbers—a system that many, but not all, humans know. In addition to young children, adults in some aboriginal tribes have been found to lack both number words and the ability to perform numeric equivalence tasks (Gordon, 2004; Frank et al., 2008; Pica et al., 2004; Pitt et al., 2022). This has sparked intrigue about the necessary conditions for humans to learn representations of exact number. Is formal education required for humans to learn about numbers? Can symbolic representations of numbers exist without possessing words for them? How did humans develop symbolic number systems in the first place?

Theories of the development of numerical abilities have often been cast within a symbolic framework, but recent evidence suggests that many of these theories may fail to capture signs of gradience in children's acquisition of number. Gelman & Gallistel (1986) proposed that children's early performance on such tasks depends on 3 symbolic counting principles: *one-to-one correspondence*, *stable ordering*, and *cardinality*. This theory was challenged, however, by Wynn (1992), who showed that children who demonstrated the ability to perform counting tasks with very small numbers often failed to perform correctly with larger values in their count lists. The idea then arose that the induction of such principles coincided with the ability perform such tasks with sets containing more than 3 or 4 items. Sarnecka & Carey (2008) offered additional tasks thought to assess reliance on these principles, but the idea that these principles applied generally to all numbers in the child's count list was not supported by the subsequent work of Davidson et al. (2012); instead these authors found evidence of gradual acquisition of the ability to perform such tasks, progressing from smaller to larger numbers within the child's count list as several number-relevant abilities improved. These findings are consistent with the view that children's numerical abilities emerge gradually, raising the possibility that their behavior may progressively align with symbolic principles as they gain more and more experience.

In our work, we take inspiration from the number cognition literature to ask if we can find symbolic representations of number in ANNs trained on numeric tasks. We then probe deeper into the relationship between model performance and symbolic alignment. We first show that we can find symbolic representations of number in an architectural variant of Recurrent Neural Networks (RNNs) known Gated Recurrent Units (GRUs) (Cho et al., 2014). We source this evidence by training a GRU to 100% accuracy on held-out test quantities in a counting task, and then we find a causal alignment between the latent representations of the trained GRU and a symbolic program. We find this alignment using a technique called Distributed Alignment Search (DAS) (Geiger et al., 2021; 2023) which enables us to test for the existence of symbolic programs in distributed representations using causal interventions. We find that the GRUs use a count up, count down strategy which increments and decrements a single count variable based on the phase of the task. We contrast this with an alternative symbolic program that solves the counting task with two separate count variables— one to track the target quantity and another to track the response quantity. We further demonstrate the significance of our results by showing that the symbolic number alignment does not emerge in models trained on a similar task that can be solved using an exact token copy operation.

We use our results to demonstrate the utility of symbolic alignments within cognitive models. The alignments in this paper demonstrate the emergence of symbolic numbers within connectionist models. These findings have implications for the origins of exact symbolic number systems and for the necessary conditions for neural systems to learn how to count. These results serve as an initial step towards the unification of the symbolic and connectionist camps of thought in cognitive modeling.

We then explore deeper into the relationship between the symbolic alignment and the model performance to show that the symbolic alignment has a strong correlation with model accuracy, often being a leading indicator of a training phase transition. The relationship between the two, however, is not a perfect one-to-one correspondence, and frequently the alignment accuracy continues to change despite the model's ceiling task performance. We relate this to the symbolic gradience observed in children's number cognition, noting the similarity of these findings to that of children learning to count. Although this work is in its early stages, it has the potential to enhance our understanding of the emergence of symbols in the human mind.

We summarize the contributions of this paper as follows:

1. We demonstrate how and why to use DAS to interpret recurrent cognitive models

2. We find the emergence of symbol-like counting variables in models trained to solve a numeric matching task, serving as a proof-of-principle for symbolic representations of number in human cognition

3. We find that the symbolic representations of number strongly co-vary with model performance, although there is not a perfect one-to-one correspondence

4. We use our findings to enhance our understanding of emergent symbolic variables in neural systems, making an early step towards unifying symbolic and connectionist frameworks

## 2 METHODS

### 2.1 OVERVIEW

At a high level, we first train a model to completion on a numeric equivalence task, and then we use Distributed Alignment Search (DAS) (Geiger et al., 2021; 2023) to test for alignments between hypothesized symbolic abstractions/programs and the model's neural representations. Specifically, DAS learns an orthogonal rotation matrix (a change-of-basis matrix) at a given model layer to align a subspace of the neural representations at that layer with a hypothesized causal variable from a hypothesized symbolic program. Concretely, we use the rotation matrix on the hidden representations, swap a subspace (some fixed number of dimensions of the rotated representations) between the rotated representations of two different model inputs. We then invert the rotation and allow the model to make predictions using the intervened representation. The training objective for the rotation matrix is to find the change-of-basis such that the model will predict the hypothesized counterfactual outputs under the intervention given the hypothesized symbolic program. To evaluate how well the NN aligns with the symbolic abstraction, we perform these causal interventions on held out data. The causal interventions enable us to verify whether or not the model is performing the task in a way that is consistent with the existence of the hypothesized symbolic abstraction.

### 2.2 TASKS

The tasks we focus on in this work consist of variable length sequences of tokens. We compare two relatively simple numeric equivalence tasks in this framework. The first task is called the *Copy Task*, and the second is called the *Counting Task*. Both tasks begin by uniformly sampling a target quantity from 1 to 20. After determining the target quantity, the sequence is constructed from two phases. The first is called the **demonstration phase** which begins with a Beginning of Sequence (BOS) token and continues with a number of demonstration tokens provided by the environment. Once the number of demonstration tokens following the BOS token is equal to the initially sampled target quantity, the environment provides a Trigger token (T) indicating the beginning of the **response phase**. The model is then tasked with outputting the same number of tokens as was observed during the demonstration phase followed by an End of Sequence (EOS) token to indicate that it has finished its response. The Copy and the Counting task variants differ as follows:

**Copy Task:** other than the BOS, T, and EOS token types, there is a single token type, C, that is used to both indicate the target quantity during the demonstration phase and the response quantity during the response phase. It is possible to solve this task without using an explicit notion of quantity, by copying the literal sequence of tokens in the demonstration sequence.

**Counting Task:** in addition to the BOS, T, and EOS token types, there are 3 demonstration token types (D1,D2, and D3) and a single response token type (RESP) that is different from the possible demonstration token types. During the demonstration phase, the demonstration tokens are uniformly sampled from the possible demonstration token types. The response phase deterministically uses the unique response token type. This task variant prevents a solution that uses a direct copy operation, and this variant has $N^3$ possible input sequences for each target quantity, $N$, making it more difficult to solve using a memorization strategy.

During DAS training we sampled 1000 sequences for training, and 500 samples of held out target quantities for validation. The held out target quantities were 4, 9, 14, and 17, selected to be relatively evenly distributed amongst the possible quantities.

### 2.3 COUNTING MODEL

We trained Gated Recurrent Units (GRUs) (Cho et al., 2014) to solve the aforementioned tasks. GRUs are a specific type of RNN that are similar to a Long Short-Term Memory (LSTM) networks except that GRUs have the advantage of using a single recurrent state vector. The hidden state, $h_t$, of the GRU at a particular step, $t$, is updated according to the following equations:

$$
\begin{aligned}
r_t &= \sigma(W_{ir}x_t + b_{ir} + W_{hr}h_t + b_{hr}) & (1) \\
z_t &= \sigma(W_{iz}x_t + b_{iz} + W_{hz}h_t + b_{hz}) & (2) \\
n_t &= \tanh(W_{in}x_t + b_{in} + r_t * (W_{hn}h_t + b_{hn})) & (3) \\
h_{t+1} &= (1 - z_t) * n_t + z_t * h_t & (4)
\end{aligned}
$$

After each update step in the sequence, we make a prediction $p_{t+1}$ of the next token, $y_{t+1}$, using softmax classification on the vector $h_{t+1}$. Concretely, we feed $h_{t+1}$—with a size of 20—into a Feed Forward Network (FFN) with a single hidden layer of 80 units, a Gaussian Error Linear Unit (GELU) nonlinearity (Hendrycks & Gimpel, 2023), dropout applied on the hidden activations with 0.5 probability to drop (Srivastava et al., 2014), and a softmax applied to the outputs of the second layer to create a probability distribution over possible output tokens. We use the cross entropy of the predictions with the ground truth tokens as the loss, $\mathcal{L}$, and minimize the loss using batch gradient decent with a batch size of 128. The loss for a batch of training data is calculated as follows:

$$
\mathcal{L}_i = -\sum_{t=1}^{S-1} y_t^\top \log(p_t) \tag{5}
$$

$$
\mathcal{L} = \frac{1}{NS} \sum_{i=1}^{N} \mathcal{L}_i \tag{6}
$$

Where $i$ refers to the index of a single sequence in the batch, $t$ is the index of the step in the sequence of length $S$ where the $0^{th}$ step is not predicted, $y_t$ is a ground truth, one-hot encoded column vector, and $p_t$ is the output of the softmax function at the end of the FFN.

We used the ground truth tokens as input at each step in the sequence during the demonstration phase (i.e., teacher-forcing) up to and including the trigger token. After the trigger token, we use the predicted token of the current time step as the input token for the next time step (i.e., autoregressive) prediction vector to select the input token for the next step.

We used PyTorch with AutoGrad (Paszke et al., 2019), an Adam optimizer (Kingma & Ba, 2017) with default settings, an initial learning rate of 0.001, a learning rate decay schedule following that described in the original transformer paper (Vaswani et al., 2017). We trained 5 model seeds, each for 300 epochs. Models were trained using Nvidia Titan X GPUs. All models achieved > 99.9% accuracy on both training and validation sequences, where accuracy refers to the proportion of responses with the correct number of RESP tokens following the T token and ending with an EOS token.

### 2.4 RECURRENT DAS

Causal abstraction is a hypothesis testing framework that manually tests if the causal mechanism of a variable in a symbolic program abstracts the causal mechanism of neural representations relative to a particular alignment (Geiger et al., 2021). Distributed Alignment Search (DAS) (Geiger et al., 2023) is a recent variant of causal abstraction where it turns the brute-force hypothesis search process into an optimization problem. DAS actively learns an aligned linear subspace of the full vector space by rotating the original neural representations using an orthonormal matrix. The question remains, how does the rotation matrix learn to find this subspace?

In this work, in each DAS training, we fix the number of dimensions of the subspace. We allow these dimensions to be contiguous within the rotated representation under the assumption that a learned rotation matrix will be able to equivalently adapt to the particular dimensions that we isolate for the intervention. This narrows the number of possible subspace dimension choices to the range 0-$D$, where $D$ is the dimensionality of the hidden state vector, $h_t$. We then perform a hyper-parameter search, trying different sizes of the subspace, and pick one based on alignment performance. For the

Table 1: Alignment Results

| Task | Program | Variable | Acc |
|------|---------|----------|-----|
| Counting | Stack | Quantity | 0.94±0.02 |
| Counting | Stack | Phase | 0.91±0.02 |
| Counting | Match | Demo Quant | 0.40±0.03 |
| Counting | Match | Resp Quant | 0.35±0.02 |
| Copy | Stack | Quantity | 0.52±0.05 |
| Copy | Stack | Phase | 0.60±0.07 |
| Copy | Match | Demo Quant | 0.24±0.02 |
| Copy | Match | Resp Quant | 0.30±0.06 |

rest of this work, we set the subspace to 10 dimensions given that the performance was relatively unchanged between approximately 5 and 15 dimensions.

With a fixed number of dimensions established, we can now write the the DAS intervention as follows:

$$h_i^v = R^{-1}(m * Rh_j^s + (1-m) * Rh_i^d) \qquad (7)$$

Where $h^s$ and $h^d$ refer to the GRU's hidden states under some randomly selected source and destination sequences respectively. $h^v$ is the resulting, intervened hidden state. The subscripts $i$ and $j$ refer to the indices of the hidden states within their respective sequences. $m$ is a mask of 1s and 0s of the same size as $h$ where $*$ denotes the Hadamard product. We use $m$ to select the neural subspace from the source vector, $h_s$, and inject it into the destination vector, $h_d$. Conceptually, we use $m$ to inject some number of units from the rotated source vector to the rotated destination vector. $R$ is the learned rotation matrix that is constrained to be orthonormal using PyTorch's orthogonal parametrization module.

The last piece of the DAS procedure is creating the training signal for the rotation matrix under the causal intervention. The general idea is to predict the counterfactual tokens under the hypothesized symbolic program given the causal intervention. See Figure 1 for a diagram of the full process. More specifically, we create the training signal by using the symbolic causal program to generate the tokens we would expect under the causal intervention. We then use these tokens for the same autoregressive, next token prediction task that we used to train the neural model on the original task. We use the model's accuracy on the counting problems to evaluate the validity of the alignment. In the case where there exists a meaningful alignment between the neural model and the symbolic program, we expect the model's accuracy to be high under the causal interventions. In the case where there does not exist a valid alignment, the resulting accuracy should be low.

In sequence based modeling, the ordering of the tokens conveys important information. For each causal intervention, we have the option of which indices to sample for both the destination and source vectors. For simplicity, we exclude indices that correspond to BOS, T, and EOS tokens. We show results that uniformly sample the source and destination indices, $j$ and $i$, from both the demonstration and response phases.

For the training, we used 10000 destination-source sequence pairs with 1000 pairs for validation. Similar to the standard model training, we used the held out target quantities 4, 9, 14, and 17 for the validation sequences. Using held out target quantities gives more insight into the generalization capabilities of the models' solutions. It also gives insight into DAS's capabilities, as DAS is a relatively young technique for interpreting neural models. We used a learning rate of 0.001 and a batch size of 512 sequence pairs. We trained the rotation matrix until convergence on the autoregressive loss.

## 2.5 CAUSAL MODELS

As previously described, DAS requires that the experimenter first has a testable, high level, symbolic, causal program. DAS then finds the best alignment of this program within the neural representation space. We tested for the existence of two different causal programs. The **Stack Program** is an

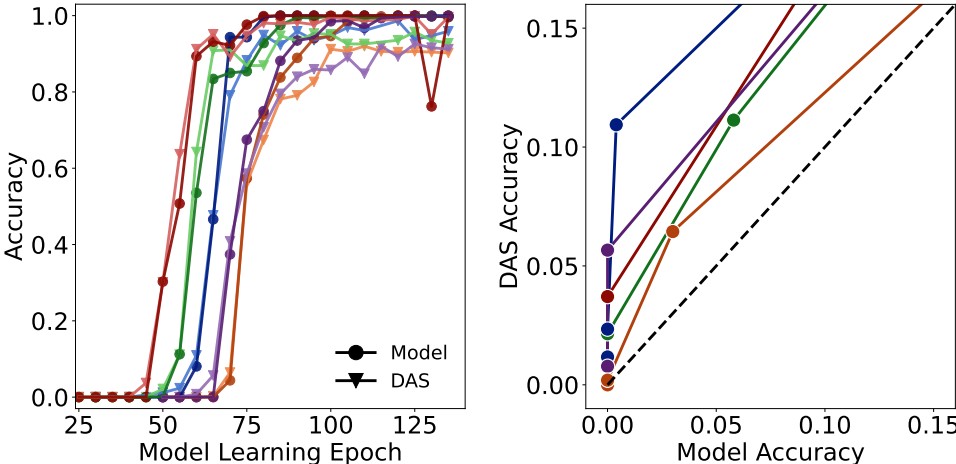

Figure 2: The relationship between model accuracy and DAS accuracy over the course of training. Each point is from a training snapshot of a single model seed trained on the Counting Task. The topmost panel shows the models' task performance in darker hues with circle markers and shows the Quantity DAS alignment performance for that same model snapshot in lighter hues with triangle markers. Here we see that the learning transition of the DAS alignment correlates with the model performance transition although they do not have a perfect correspondence. In the lower panel, the x-axis shows the model performance on the task at that snapshot with the DAS alignment performance for that same snapshot on the y-axis. Each color in the lower panel corresponds to a different model seed. We see from the lower panel that DAS performance appears to be a leading indicator of the model's learning transition.

algorithm in which the model uses a Quantity variable ($Q$ in Figure 1) to track the count and a Phase variable ($P$ in Figure 1) to track the phase of the sequence. It increments the Quantity during the demonstration phase and decrements it down during the response phase, knowing to stop when it hits 0. We included Algorithms 1 and 2 in the Appendix to detail a step-by-step account of a single sequence step in the the Stack program. In contrast, the **Match Program** uses a Phase variable and two count variables—Demo Quant and Resp Quant. It increments these count variables during the demonstration and response phase respectively knowing to stop when Demo Quant is equal to Resp Quant during the response phase.

We note the existence of an infinite number of equivalent implementations of each of these programs respectively. For example, an equivalent program to Stack is one where the program immediately adds and subtracts 1 from the Quantity variable before carrying out the rest of the program as we have described. DAS only has the ability to discriminate between programs that are causally distinct from one another. We wish to acknowledge that there are many more experiments that we could do to further refine the symbolic program(s) in this work. We leave this exploration to future work.

## 3 RESULTS/DISCUSSION

We begin by addressing the symbol-like nature of the NN task solutions and by demonstrating that DAS can help us identify which symbolic program the NNs find. We can see from Table 1 the results of our DAS alignments. First, we note that we see relatively high alignment accuracies in the Quantity and Phase variables from the Stack program for the Counting Task models compared to the results from the Match program. We see this comparative difference when comparing the Quantity variable to both the Demo Quant and Resp Quant variables. These comparisons suggest that the Stack model characterizes the solutions better than the Match program.

To further explore the meaning of our results, we compare models trained on the Copy and Counting tasks. The purpose of this comparison is to demonstrate that our findings for the Counting Task models are not merely performing an object dependent, copy operation, but rather performing an object

independent numeric operation. We see from the comparatively low alignments of the Copy Task models that the two training variants have resulted in networks with different solutions. Although we cannot make claims about the exact nature of the model solutions from this result, we can use it to further delineate the interpretation of our alignment accuracies.

These results demonstrate that symbol-like representations of discrete numbers do emerge in neural models that have only been trained to produce the same number of items that they first observed. This is a proof of principle that neural systems do not need formal instruction for symbolic representations of number to emerge, nor do they need built in counting principles to inform their learning. This has implications for the mechanisms by which humans developed symbolic number systems. Perhaps some cultures placed a high value on precisely equal division of resources, or some similar task, creating situations that encouraged the ability to represent exact quantities.

We turn now to the developmental trajectory of the causal alignments displayed in Figure 2. We can see from the performance curves in the upper panel that both the model accuracy and DAS performance begin a transition away from 0% at similar epochs. They also exhibit similar performance trajectories. We note that in all cases, the jump in DAS alignment accuracy precedes that of the model, as shown in the bottom panel of Figure 2. This result can be contrasted with a hypothetical result in which the alignment curves lag behind the performance of the model. In this hypothetical, alternative case, a possible interpretation could have been that the network develops unique solutions for many different input-output pairs and progressively unifies these solutions. The results that we see in Figure 2 can be used as evidence for an early sign of an emergent, symbol-like solution in the NNs even at earlier training epochs. Perhaps this is to be expected in light of works like Saxe et al. (2019) and Saxe et al. (2022) which show an inherent tendency for NNs to find solutions that share network weights. Perhaps, also, these results would be different for a non-linear task.

Despite the similarities between the model performance and alignment performance in Figure 2, we can see that the alignment performance often fails to achieve 100% alignment. Furthermore, the alignment performance continues to vary despite the models' ceiling task performance. We interpret these results as a reminder that representations in distributed systems exist on a continuum despite seemingly discrete, symbolic performance on the task. These results have an analogy to children's number cognition in which children may appear to possess a symbol-like understanding of exact numbers and their associated principles, but when probed deeper, the symbol-like picture falls apart. This can also be related to the Large Language Model (LLM) literature, in which the notion of sharp changes in performance as a function of LLM scale have been demonstrated to be a function of metric rather than a sudden change in innate ability Schaeffer et al. (2023).

We use these results to highlight the nuances of symbolic interpretations in cognitive science. In one light, we have shown that we can find causal symbols within the distributed representations of the neural systems. We have also shown that these symbol-like representations emerge in a way that correlates with model performance. These results can inform our thoughts on the nature of distributed solutions in cognitive models—symbol-like representations are perhaps an inherent property of general distributed solutions, at least when the task is one that can be solved by relying on such representations. In another light, however, we see that it is easy to overly simplify our understanding of a distributed system based on task performance alone. There is a nuanced picture to distributed solutions that must be considered in order to understand the full picture of human cognition.

Lastly, we highlight the utility of DAS for interpreting distributed systems. In this work, DAS has revealed an important piece of the puzzle to understanding number representations in neural systems. DAS also demonstrated the subtle, graded nature of distributed solutions, which we see in human cognition. These findings serve as a bridge between connectionist and symbolic frameworks for understanding human cognition, suggesting that DAS will continue to be a useful tool for understanding the computations performed in cognitive models that rely on ANN systems.

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

## A    APPENDIX

---

**Algorithm 1** One sequence step of the Stack Program

---

$q \leftarrow Quantity, \ p \leftarrow Phase, \ y \leftarrow$ input token
**if** $y ==$ BOS **then**                    ▷ BOS is beginning of sequence token
    $q \leftarrow 0, p \leftarrow 0$
    return sample(DEMO)                    ▷ sample a demo token
**else if** $y \in$ DEMO **then**              ▷ DEMO is set of demo tokens
    $q \leftarrow q + 1$
    return sample(DEMO)
**else if** $y ==$ TRIG **then**               ▷ TRIG is trigger token
    $p \leftarrow 1$
**else if** $y ==$ RESP **then**               ▷ RESP is response token
    $q \leftarrow q - 1$
**end if**
**if** $(q == 0) \ \& \ (p == 1)$ **then**
    return EOS                            ▷ EOS is end of sequence token
**end if**
return RESP

---

**Algorithm 2** One sequence step of the Match Program

---

$d \leftarrow DemoQuant, r \leftarrow RespQuant,\ p \leftarrow Phase,\ y \leftarrow$ input token
**if** $y ==$ BOS **then**                                      ▷ BOS is beginning of sequence token
    $d \leftarrow 0, r \leftarrow 0, p \leftarrow 0$
    return sample(DEMO)                                       ▷ sample a demo token
**else if** $y \in$ DEMO **then**                              ▷ DEMO is set of demo tokens
    $d \leftarrow d + 1$
    return sample(DEMO)
**else if** $y ==$ TRIG **then**                               ▷ TRIG is trigger token
    $p \leftarrow 1$
**else if** $y ==$ RESP **then**                              ▷ RESP is response token
    $r \leftarrow r + 1$
**end if**
**if** $(d == r)\ \&\ (p == 1)$ **then**
    return EOS                                                ▷ EOS is end of sequence token
**end if**
return RESP

---

