# OpenReview forum: "Symbolic Variables in Distributed Networks that Count"
_ICLR.cc/2024/Workshop/Re-Align — ICLR 2024 Workshop Re-Align Poster_

### Official Review · Reviewer_ZCgJ · 2024-02-23
**Counting in Neural Networks**

**Rating:** 2
**Fit:** 3
**Confidence:** 2

**Workshop Review:**

Clarity: I thought the discussion of number-sense in humans was very clear (if a bit verbose considering the limited ability to later make correspondence between the presented results and the interesting previous literature). I thought the discussion of DAS was confusing. DAS was introduced three different times, the first at a very high level (fair enough), then later at two further detailed levels. I understood what the method was morally trying to do, but had to spend a long time decoding its mechanics from these two descriptions. Section 2.4 specifically could do with some cleaning up.

I ended up concluding the authors were learning an orthogonal matrix so that the first D dimensions of the hidden representation encoded the thing closest to the meaningful cognitive variable as possible. This rotation was learnt by requiring that injecting the neural activity from one model's top D dimensions changes the second model's outputs in just the way that you would predict were the neural network actually just implementing the proposed symbolic algorithm. (Writing this I see it is hard to explain ... ;). Hopefully that is right, if not may it serve as one example of stupid people reading the paper and getting confused, which may help in adding more clarity to the writing.

Correctness: All the details appeared correct, and the model descriptions were thorough (barring complaints about DAS above).

Novelty: The work was interesting, though, as the authors point out, definitely had an ongoing feeling. The results in table 1 had a cool separation between the two tasks, and between stack and match, and I found the right panel of figure 2 interesting.

That said, I wondered about one aspect of the discussion of symbolic reasoning. If your proposed symbolic mapping does not reach 100% is that because the neural network reasoning is non-symbolic or because the proposed algorithm is slightly wrong?

Interest to community: definitely of interest, and I look forward to seeing this work's development.

**Reason For Not Giving Higher Score:**

Because it did not feel like a finished piece of work, it felt like some cool preliminary results that warrant further exploration.

**Reason For Not Giving Lower Score:**

Because, though preliminary, the results were cool! Further, the paper was well put together.

**Reviewer Domain:**

neuroscience

---

### Official Review · Reviewer_wicr · 2024-02-24
**Review of Submission 34**

**Rating:** 2
**Fit:** 3
**Confidence:** 2

**Workshop Review:**

In this paper, the authors study symbolic explanations of Recurrent Neural Networks (RNNs) in solving counting and copying tasks. The authors investigate the degree of alignment of trained models to symbolic algorithms via Distributed Alignment Search (DAS). The experimental setting is novel (with the use of DAS) and, while the results are somehow preliminary, the problem of how number sense can emerge in RNNs is an interesting interdisciplinary topic.

This work fits the criteria for acceptance of the workshop, being well-written and proposing a simple, yet worth-exploring research. Other comments are shown below.

**Reason For Not Giving Higher Score:**

Some parts of the paper can be improved. One is the writing of some parts, whose claims are not entirely supported by the evidence. For example, I could not make much sense of what the authors mean "a crucial step towards unifying symbolic and connectionist frameworks". Bridging between symbolic and connectionist models is very interesting, but it is the case that the results hold some approximations: the problem is inherently symbolic (counting and copying) and DAS makes use of linearity assumptions. Moreover, the results in some respect confirm those already obtained by Geiger et al. (Neurips 2023) - Interpretability at scale, so it should be clear on what aspect the authors are differing by already known results.

Also, Figure 2 can be improved. The overlap between "model" and "DAS" makes it difficult to see the difference between the two scores. The results also support the fact that counting task accuracy and DAS show similar behavior (the correlation seems very high). This could be further explored, as well as other tasks and other algorithms. Concluding, I am not sure that the gap between DAS and accuracy at the early stages of training is a sensible signal for an "emergent, symbol-like solution in the NN". This could be further investigated in the current setting.

**Reason For Not Giving Lower Score:**

The paper is well written and clear in the presented material. The scope is interesting for the audience at the workshop and performs a study with a setup, via DAS, that is very relevant for alignment research. While the results are preliminary, the work is investigating a promising direction.
I would suggest the authors connect to other explainability works regarding global explanations and concept-level explanations. Also, The Clock and the Pizza: Two Stories in Mechanistic Explanation of Neural Networks, Zhong et al (NeurIPS 2023) is very related to their analysis.

**Reviewer Domain:**

machine learning

---

### Official Review · Reviewer_dfqL · 2024-02-25
**Nice DAS example, more explanation would help**

**Rating:** 3
**Fit:** 3
**Confidence:** 2

**Workshop Review:**

This paper demonstrates the use of a causal interpretability method, DAS, in a small GRU RNN trained to perform copying and counting generation tasks. The work shows that DAS can compare several symbolic algorithms to a neural network to determine the extent to which any of them align with the network's internal computations. This line of work is important, relevant, and aligns (ahem) with the theme of the workshop.

My main suggestions are as follows:
- Provide a brief justification of the system you choose to study.
- How much would a method need to be adapted from GRU to other systems? You mention that GRUs conveniently only have one state vector - would RNNs that have more be harder to study? What about feedforward systems like transformers? In general, what are the criteria that make a distributed system suitable or unsuitable for DAS? [I know this is not the focus of this paper but a brief comment would be helpful for framing]
- The representational link from the symbolic algorithm to the neural net's representations is critical and perhaps the biggest assumption of the method. Could you justify your linking function choice (a linear subspace) and discuss its possible limitations?
- Briefly review past literature on internal mechanisms of counting in RNNs. Do not have to be these ones but maybe:
Weiss, G., Goldberg, Y., & Yahav, E. (2018). On the practical computational power of finite precision RNNs for language recognition. arXiv preprint arXiv:1805.04908.
El-Naggar, N., Madhyastha, P., & Weyde, T. (2022). Exploring the Long-Term Generalization of Counting Behavior in RNNs. arXiv preprint arXiv:2211.16429.

Minor:
Consider moving Figure 1 to the point of first mention; it comes a bit too early in the current paper

**Reason For Not Giving Higher Score:**

NA

**Reason For Not Giving Lower Score:**

Relevant, timely, of interest to the workshop audience

**Reviewer Domain:**

neuroscience

---

### Decision · Program_Chairs · 2024-03-02

Accept (Poster)